# Efficient and Rapid Removal of Pb(II) and Cu(II) Heavy Metals from Aqueous Solutions by MgO Nanorods

**Monira G. Ghoniem** [1,*] [ID]**, Mohamed Ali Ben Aissa** [2] [ID]**, Fatima Adam Mohamed Ali** [1] **and Mohamed Khairy** [1,3,*]

1   Chemistry Department, College of Science, Imam Mohammad Ibn Saud Islamic University (IMSIU), Riyadh 11623, Saudi Arabia
2   Department of Chemistry, College of Science and Arts, Qassim University, Ar Rass 51921, Saudi Arabia
3   Chemistry Department, Faculty of Science, Benha University, Benha 13518, Egypt
*   Correspondence: mgghoniem@imamu.edu.sa (M.G.G.); mohkhairy@fsc.bu.edu.eg (M.K.);
    Tel.: +966-5-4053-9950 (M.G.G.); +20-1-0710-6578 (M.K.)

**Abstract:** In this study, the adsorption capability of MgO nanorods for the quick and effective elimination of Cu(II) and Pb(II) heavy metals from wastewater was examined. The MgO nanorods were produced via simple coprecipitation process. Various characterization techniques were used to investigate the morphological and chemical properties of the as-prepared nanomaterial. Moreover, the influences of initial heavy-metal ion concentration, pH, and contact time were investigated to evaluate the removal efficiency of the nanomaterials. The adsorption process followed pseudo-second order and Langmuir adsorption isotherm models, according to kinetics and isotherm investigations, respectively. MgO nanoparticles exhibited a high adsorption capacity for Cu(II) (234.34 mg/g) and Pb(II) (221.26 mg/g). The existence of interfering ions in the aqueous solution leads to a decrease in the adsorption capacity. Surface complexation was determined as the key contributor to the adsorption of Cu(II) and Pb(II) heavy-metal ions onto MgO nanorods. Notably, regeneration experiments demonstrate the potential applicability of MgO nanorods for the elimination of Pb(II) and Cu(II) from aqueous solution.

**Keywords:** MgO nanorods; Cu(II) and Pb(II) ions removal; adsorption mechanism; surface complexation

## 1. Introduction

The collective effect of industrial waste release, polluted with huge amounts of toxic dyes, heavy metal ions, and other pollutants from different world industries, poses many environmental challenges. They are nonbiodegradable, so their growth will have a negative impact on human health [1–4]. The World Health Organization has optimized the maximum level of these environmental pollutants [5]. By exceeding this level, human life will certainly be undesirably affected.

Heavy metals are elements that are less noxious and have high density. Examples of heavy metals are copper, lead, mercury, cadmium, arsenic, and chromium, and the density of these heavy metals is more than 6 g/m$^3$ [6]. Lead and copper are toxic elements for human, animal, and plant health [7,8]. They are the main contaminants in drinking water, via human sources, herbal plants, mining, and industrial wastes [9–15]. Lead is a carcinogenic metal, but copper is toxic; large doses of copper can be very hazardous and chronic, causing several health diseases when consumed, such as diarrhea, irritated skin, vomiting, nausea, and abdominal illnesses [7,8,15]. The allowed limit cited by the World Health Organization and the European Union for copper and lead in drinking water are 2 and 0.01 ppm, respectively [16]. Moreover, the maximum limit for copper and lead in wastewater is 1 mg/L as cited by the South African National Standard [17]. Therefore, there is an essential need to design innovative maintainable materials to eliminate these pollutants [7,8]. The research areas for the significant detection of heavy metal ions (HMIs) from the aqueous phases have become very important. As a result, increasing efforts are being

made to develop an approved technology for wastewater treatment. Chemical and physical procedures such as chemical oxidation [18], ozonation, adsorption [19], froth flotations, reverse osmosis, membrane separation [20], coagulation [21], flocculation, and electro-coagulation are often utilized in the treatment process [19]. Regarding dye elimination, adsorption is the favored technique between the others used because of its affordable price, easy preparation, and a diminutive amount of energy consumption [22–24]. Therefore, developing reliable adsorbents nowadays becomes a necessity

Nanoparticles (NPs) have received much attention, since they are harmless and exhibit unique features at the nanoscale that improve their potential for use in various disciplines [25–28]. NPs have recently been used in various fields, including cosmetics, the textile industry, pharmaceuticals, medicine, agriculture, photocatalytic activity or biodegradation, and wastewater treatment processes [29–33].

Regarding heavy metal ion sensing, many adsorbents were used, such as metal–organic frameworks [34], carbon-based materials [35], inorganic–organic hybrid materials [36], and metal oxides [9]. To maintain thermal stability and structural order, the inorganic part of the nanocomposite plays an essential role due to its physicochemical properties [37]. Many searches were performed on the removal of Pb(II) and Cu(II) ions on various adsorbents such as biomaterials with adsorption capacity for Cu(II) (6.707 mg/g) and Pb(II) (18.79 mg/g) [15], chelating resins [38], and nanostructured lithium, sodium and potassium titanates [39].

Magnesium oxide nanoparticles (MgO-NPs) have a wide range of applications, including semiconductors, cell imaging, and in the paint industry [40–42]. Moreover, MgO-NPs are an atypical adsorbent that are utilized in wastewater and heavy metal treatment processes due to their eco-friendliness, low cost, and high level of safety [43,44]. In addition, the high surface activity, biocompatibility, high stability, and enhanced active sites on their surfaces are responsible for these uses in wastewater treatment [40,41,45,46]. In addition, the activities are associated with features such as shape, size, and fabrication technique [47]. Numerous physical and chemical methods, including solvo/hydrothermal, co-precipitation, combustion, sol-gel, and green synthesis, can be used to produce MgO NPs [40,42,48].

At this point, the MgO nanorods have been synthesized by simple precipitation method and the effect of Pb(II) and Cu(II) ions adsorption has been studied. Our results reveal that the nanomaterial shows an outstanding high adsorption ability of Pb(II) and Cu(II) ions. In addition to, the adsorption mechanism, the kinetics and isotherms have also been explored.

## 2. Experimental

### 2.1. Chemicals

Ammonia solution ($NH_4OH$, 25%), copper (II) sulfate pentahydrate ($CuSO_4.5H_2O$, $\geq$99%), hydrochloric acid (HCl, 37%), lead (II) sulfate ($PbSO_4$, 98%), sodium chloride (NaCl, $\geq$99%), sodium nitrate ($NaNO_3$, $\geq$99.5%), potassium nitrate ($KNO_3$, $\geq$99.0%), sodium hydroxide (NaOH, $\geq$99%), and magnesium nitrate ($Mg(NO_3)_2$, $\geq$99.0%), were bought from the Merck Company and employed without additional purification. The Pb(II) and Cu(II) required concentrations of (5 to 200 ppm) were obtained by diluting the Cu(II) and Pb(II) stock solution (1000 ppm).

### 2.2. MgO Nanorods Manufacturing

MgO nanorods were produced in two steps: precipitation of $Mg(OH)_2$, followed by calcination at 750 °C until MgO nanorods fabrication. $Mg(OH)_2$ was produced by the gradual addition of $NH_4OH$ to 0.2L of $Mg(NO_3)_2$ solution (0.1 M) until precipitation, followed by 12 h of stirring with a magnetic stirrer. After washing, the precipitate was dried and calcinated at 750 °C to obtain magnesium oxide nanorods.

### 2.3. Characterization Methods

X-ray diffraction (XRD) was performed using a Rigaku Mini Flex 600 (Matsubara-cho Akishima-shi, Tokyo 196-8666, Japan) diffractometer that was outfitted with a CuK radiation source (λ = 1.5417 Å) in order to determine the crystal structure and phase purity of the substance. The FTIR spectra of MgO nanorods after and before heavy metals adsorption were investigated by a JAS-CO FT-IR spectrometer (28600 Mary's Court, Easton, MD 21601, USA). The MgO's porosity and surface area were determined using the Brunauer-EmmettTeller (BET) (4356 Communications Drive, Norcross, GA 30093, USA) formula and the t-plot method of Lippens and de Boer. Before experimenting, the sample was outgassed under He gas flow at 200 °C for 2 h to remove moisture and sorbed contaminants. The morphology of the as-obtained MgO nanoparticles was analyzed by field emission scanning electron microscopy (FE-SEM) (JEOL USA, Inc. 11 Dearborn Road, Peabody, MA 01960, USA) and transmission electron microscopy (TEM). This was performed with a high-resolution JEOL JEM-6700F (Suite 320 Santa Clara, CA 95054, USA) instrument coupled with electron dispersive X-ray spectroscopy (EDS) to determine the elemental chemical composition of the nanoparticles and TEM by JEOL-2010 (7431 NE Evergreen Parkway, Suite 130, Hillsboro, OR 97124, USA). The sample adhered on double-sided carbon conductive tape with adhesive using a clean sample holder and forceps. The Pb(II) and Cu(II) ions concentration was measured using an ICP Spectro Genesis spectrometer (No. 2, Jalan Jururancang U1/21, Hicom Glenmarie Industrial Park Seksyen U1, 40150 Shah Alam Selangor Malaysia).

### 2.4. Adsorption Experiments Details

Under electromagnetic stirring for 24 h, adsorption studies were performed in a glass vial containing 25 mL of Cu(II) and Pb(II) ions mixture at various initial concentrations and 10 mg of MgO nanoparticles. After adsorption equilibrium, the mixture was centrifuged to determine the residual concentration of Cu(II) and Pu (II) ions using ICP. The adsorption capacity of Cu(II) and Pb(II) (Qe, mg/g) were computed using Equation (1) [14]:

$$Q_e = \frac{V}{w}(C_i - C_e) \tag{1}$$

where $Q_e$ is the adsorption capability (adsorption of the dye per unit mass of the sample, mg g$^{-1}$), $C_i$ is the initial concentration and $C_e$ is the equilibrium concentration of the ions in the solution (mg L$^{-1}$), W is the amount of adsorbent (g), and V is the solution volume (L).

For the kinetic experiment, the volume was 250 mL, the initial Cu(II) and Pb(II) concentrations were 100 ppm, and the mass of the MgO nanoparticles was 100 mg. The evaluation was conducted in complete darkness and involved magnetic stirring. The remaining Pb(II) and Cu(II) concentrations were calculated by removing 5 mL of the suspension and centrifuging at predetermined time intervals. The adsorption capacity of Cu(II) and Pb(II) at time t were computed using Equation (2):

$$Q_t = \frac{V}{w}(C_i - C_t) \tag{2}$$

where $Q_t$ is adsorption capacity of Cu(II) and Pb(II) at time t (mg/g), $C_i$ is the initial concentration and $C_t$ is the concentration of the ions in the solution at time t (mg/L), W is the amount of adsorbent (g), and V is the solution volume (L).

For the pH-effect experiments, the Cu(II) and Pb(II) ions concentrations with an initial pH were set in solution at various pH values ranging from 3 to 7 by adding NaOH (0.05 mol/L) or HCl (0.05 mol/L)

For the regeneration study, the MgO nanorods were collected after the adsorption experiment using a centrifuge and rinsed with distilled water. The desorption process was accomplished by combining the recovered MgO nanorods with 50 mL of NaOH (0.1 M) solution for 30 min with magnetic stirring. The precipitate was then thoroughly washed

and dried at 100 °C for two hours before being employed to eliminate Pb(II) and Cu(II) ions.

## 3. Results and Discussion

### 3.1. MgO Nanoparticles Characterization

The X-ray diffraction pattern was used to estimate the structural phase of the as-obtained MgO nanorods by the co-precipitation method. The XRD pattern of the as-obtained MgO is shown in Figure 1a. This pattern reveals that the main diffraction peaks at 2θ = 37.1°, 43.0°, 62.3°, 74.8°, and 78.7° could be likened to the (111), (200), (220), (311), and (222) planes (JCPDS 87-0653) [49], which suggests the creation of polycrystalline cubic structures of MgO nanoparticles. Additionally, no distinctive peaks of additional contaminants were seen, demonstrating the product's excellent purity. Furthermore, the sharp (narrow width) and strong (high intensity) diffraction peaks suggest that the obtained MgO NPs were of good crystallinity. The Debye–Scherrer formula was employed to determine the crystallite size of the MgO NPs based on their (200) lattice plane [50].

$$D = \frac{k\,\lambda}{\beta\cos\theta} \tag{3}$$

where λ is the wavelength of the employed X-rays (1.54056 Å), β is the full width half maximum height of the diffraction peak at an angle θ and K is a constant equal to 0.9. The crystallite size calculated was found to be 18 nm.

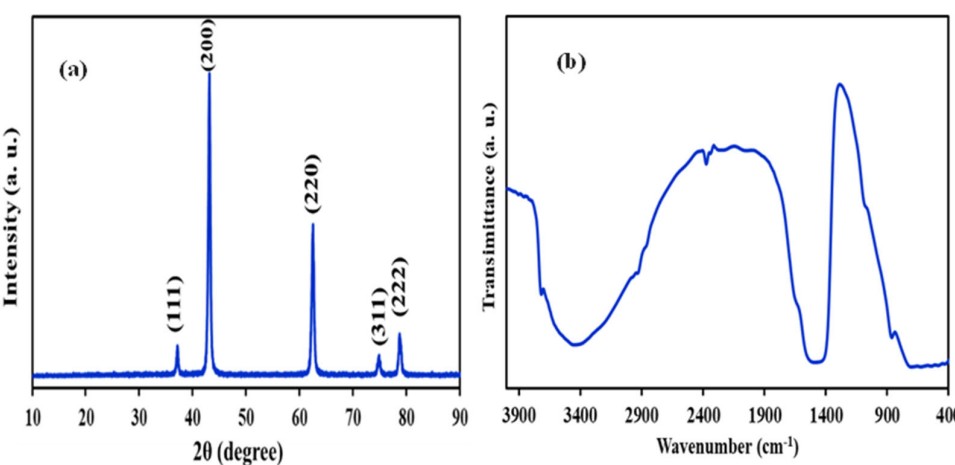

**Figure 1.** (**a**) XRD patterns, and (**b**) FTIR spectrum of MgO nanorods.

The FTIR spectrum of the as-fabricated MgO nanorods are shown in Figure 1b. The stretching vibration of the O–H group was attributed to the bandwidth between 3200 and 3700 cm$^{-1}$ [51]. Bands between 1455 and 1645 cm$^{-1}$ were connected to the –OH stretching mode of the adsorbed water molecule. The presence of Mg-O bands was confirmed by the distinctive bond at 863 cm$^{-1}$ [52].

Figure 2 shows the pore size distribution and the N$_2$ adsorption–desorption isotherms of the as-synthesized MgO nanoparticles. It was found that the isotherm is a type IV characteristic to mesoporous structure. It was observed that the hysteresis loop belongs to the category H3 type of pores, which do not display limiting adsorption at high P/P$^0$. The H3 type is formed due to aggregation of plate-like particles giving rise to slit-shaped pores [53,54]. MgO nanoparticles' BET surface area was found to be 12.2 m$^2$/g. The fact that the loop closes at a comparatively greater pressure (P/P$^0$ = 0.8–1) indicates that the mesopores have grown in size. This is supported by the pore size distribution curve generated by the Barrett–Joyner–Halenda technique, which demonstrates the developed bimodal type of pores with an average pore diameter between 1.7 and 4 extended to 700 Å average pore diameter.

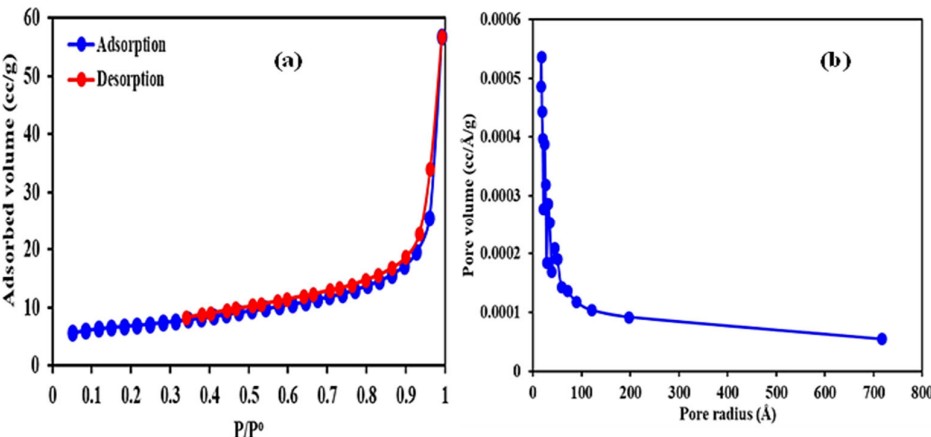

**Figure 2.** (**a**) $N_2$ adsorption–desorption isotherm of MgO nanorods and (**b**) Pore size distribution.

SEM micrographs are displayed in Figure 3a to demonstrate the morphological and structural characteristics of the produced MgO nanoparticles. MgO seems to have a homogeneous structure resembling a rod. EDS and SEM were used to determine the elemental mapping and elemental composition of the generated MgO (Figure 3b). The EDX spectrum of MgO also confirmed the attendance of a sharp and strong Mg peak in addition to the oxygen peak. The EDX spectrum also recorded the presence of carbon (C), which may be generated from double-sided carbon conductive tape. The generated nanomaterials' elemental mapping showed that MgO nanorods consists of O and Mg elements, (Figure 3d,e), suggesting the formation of MgO nanorods. On the other hand, transmission electron microscopic (TEM) micrograph (Figure 3f) of the prepared MgO confirm the formation of nanorods of MgO with average diameter 10.7 nm and length of 156 nm with high homogeneity.

### 3.2. Cu and Pb Ions Adsorption Study

### 3.2.1. Effect of Initial Pb(II) and Cu(II) Concentrations

The effect of initial Pb(II) and Cu(II) concentrations on the adsorption performance of MgO nanorods was investigated under the following operating conditions: 24 h of contact time, room temperature, a pH value equal to 3, and a 10 mg of nanosorbent dose. As displayed in Figure 4a, increasing the initial Pb(II) and Cu(II) concentrations from 5 to 75 mg/L improved the removal percentage significantly from 83.96 to 98.98% for Cu(II) ion and from 47.47 to 95.22% for Pb(II) ion.

### 3.2.2. Effect of pH on Pb(II) and Cu(II) Removal

It is well known that the pH value is essential for comprehending the surface interactions between MgO nanorods and Pb(II) and Cu(II) adsorbates ions. The impact of pH on Pb(II) and Cu(II) adsorption capacity was studied over the pH range of 3.0 to 7.0, with the optimal adsorption capacity obtained for Cu(II) and Pb(II) ions at pH value equal to 3, as depicted in Figure 4b. Important protonation appears on the adsorbent surface at low pH values, resulting in weakly adsorbent–ions interactions. Analogous results were found in reported studies [11,55]. The solubility of metal ions such as Pb(II) and Cu(II) depends on the pH value. Pb(II) and Cu(II) are extremely soluble as Pb(II) and Cu(II) free ions and as $Cu(OH)^+$ and $Pb(OH)^+$ at lower pH values [48,56]. In addition, Pb(II) and Cu(II) ions will precipitate as metal hydroxide $Pb(OH)_2$ and $Cu (OH)_2$ at pH values greater than 7 [57,58].

### 3.2.3. Effect of Interfering Ions on Pb(II) and Cu(II) Removal

The effect of interfering Na(I) and K(I) ions on the adsorption of Cu(II) and Pb(II) onto MgO nanorods were examined. To study the influence of interfering ions on potential Cu(II) and Pb(II) ion adsorption, the initial concentration of Cu(II) and Pb(II) was set as 100 ppm, the pH of the solution was 3, and the concentration of interfering ions was 0.005 M. Figure 5

depicts the influence of interfering ions on the adsorption capacity of Cu(II) and Pb(II) onto MgO nanorods. As shown, the tendency of potassium ions to compete for the active site was significantly higher than sodium ions. This could be due to the difference in the hydrated ionic radii of sodium (2.76 Å) and potassium (2.32 Å). Thus, as the cation's ionic radius decreases, the influence of competition on the binding sites increases. Consequently, potassium ions are more influential than sodium ions. The results are consistent with Musumba et al.'s findings [59].

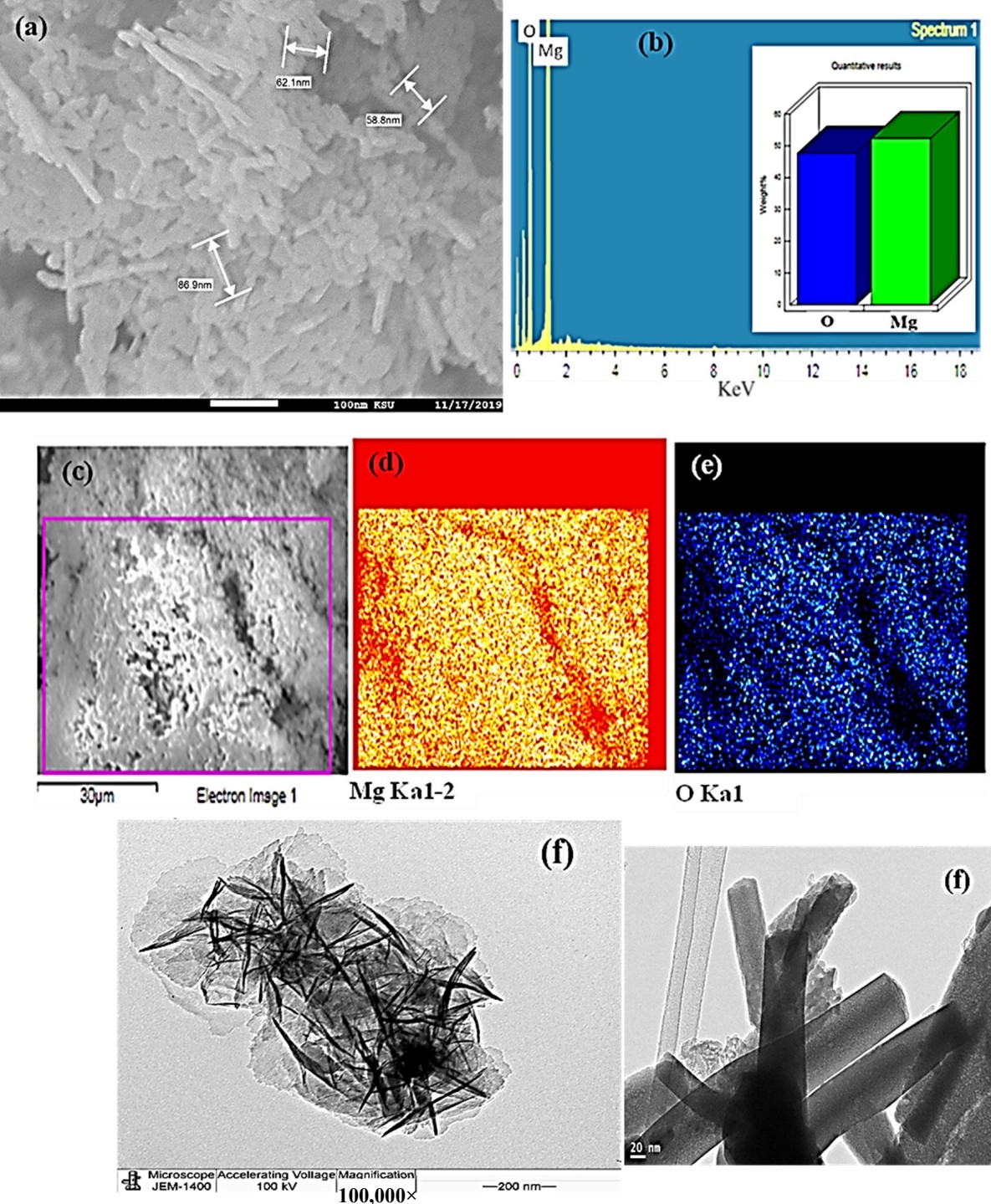

**Figure 3.** SEM image (**a**), EDX spectrum of MgO (**b**); and X-ray elemental mapping of Mg−Ka (**d**) and O−Ka (**e**) of the SEM MgO selected area (**c**), and TEM image (**f**).

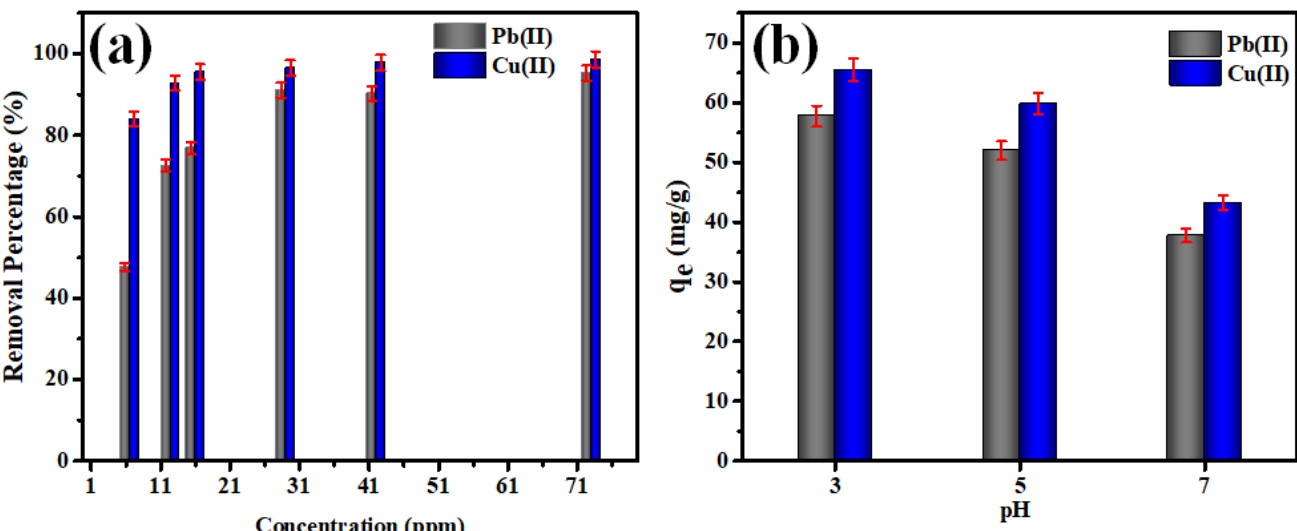

**Figure 4.** (**a**) Impact of initial Pb(II) and Cu(II) concentration on the adsorption onto MgO nanorods at pH = 3, (**b**) the effect of pH on % elimination of Pb(II) and Cu(II) with an initial concentration of 45 ppm.

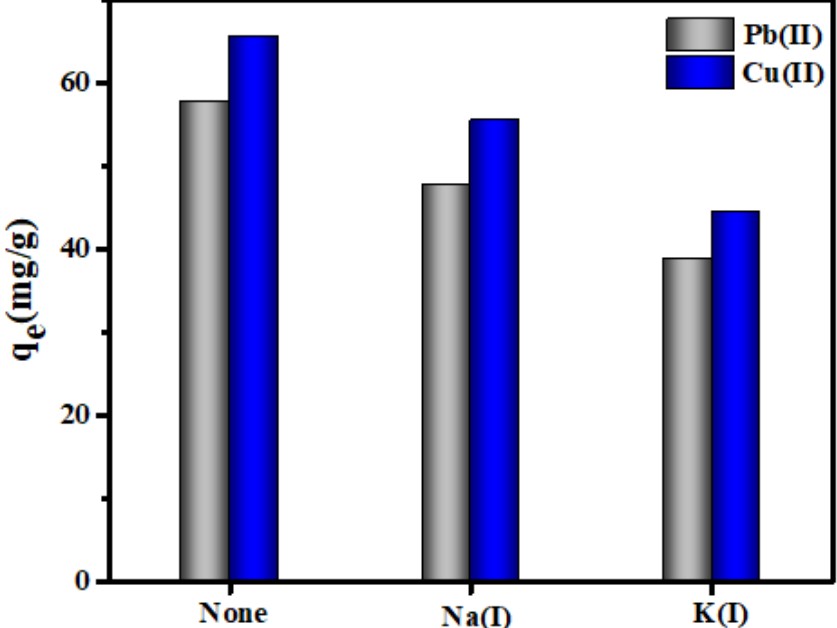

**Figure 5.** Influence of interfering ions on the adsorption of on Pb(II) and Cu(II) ions using MgO nanorods.

### *3.3. Adsorption Kinetics*

#### 3.3.1. Equilibrium Contacts Time

Figure 6a illustrates the effect of contact time on Pb(II) and Cu(II) ion elimination. At an initial ions concentration of 45 ppm, the adsorption of Pb(II) and Cu(II) ions onto MgO nanorods was studied during a stirring time range of 5 to 1440 min. As depicted in Figure 6a, the removal rate was remarkably rapid during the first few minutes of contact time (17 min for Cu(II) and 24 min for Pb(II)), and then gradually decreased after the equilibrium was reached. The amount of Pb(II) and Cu(II) ions adsorbed reaches approximately 116 mg/g and 123 mg/g, respectively, during a short time interval. An adsorbent with a wide surface area and porosity is projected to expose many adsorption sites, resulting in a significant adsorption capacity.

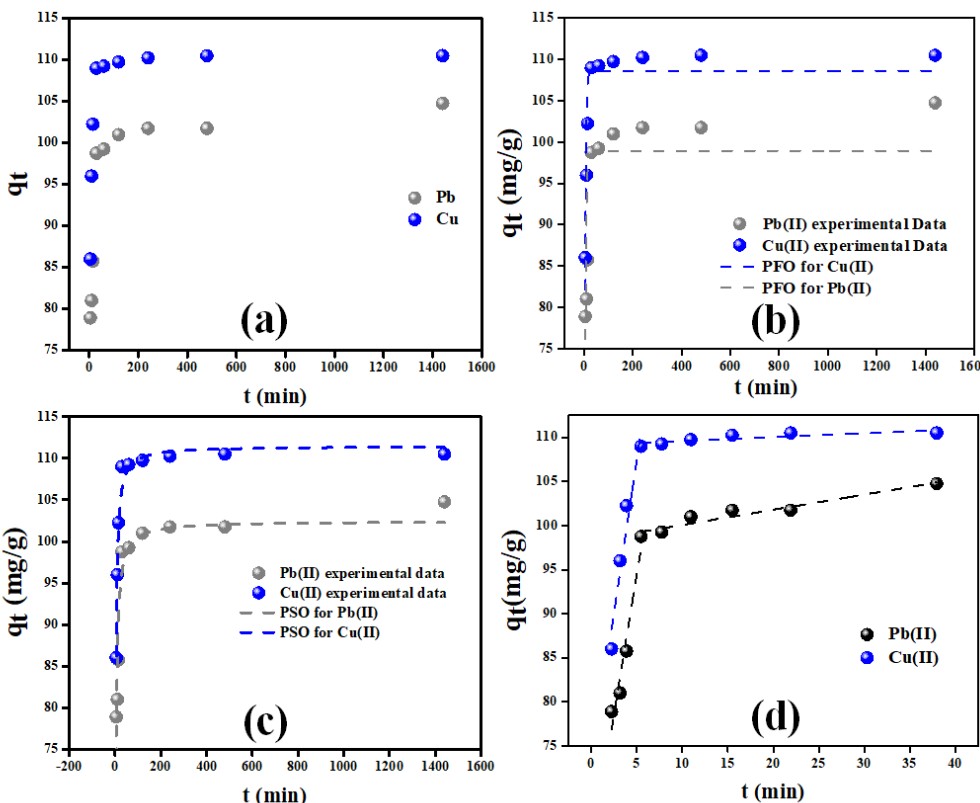

**Figure 6.** (**a**) Contact time (**b**) PFO, (**c**) PSO, and (**d**) intra–particle-diffusion on the adsorption of $Pb^{2+}$ and Cu(II) onto MgO nanorods.

### 3.3.2. Adsorption Kinetic Studies

Adsorption kinetics parameters are essential for predicting the effectiveness of an adsorbent. As given in Figure 6, the adsorption of Pb (II) and Cu (II) ions onto MgO nanorods was evaluated using pseudo-second-order (PSO), pseudo-first-order (PFO), and intra-particle diffusion (IPD) models. The corresponding nonlinear mathematical equations are presented in Table 1.

**Table 1.** Kinetics models parameters for the adsorption of Pb(II) and Cu(II) ions by MgO nanorods.

| Kinetics Models | Equation N° | Parameters | Cu(II) | Pb(II) |
|---|---|---|---|---|
| PFO | 4 | $q_e$ (mg g$^{-1}$) | 108.61 | 98.93 |
| | | $k_1$ (min$^{-1}$) | 0.28 | 0.25 |
| | | $R^2$ | 0.852 | 0.575 |
| PSO | 5 | $q_{e\,(calculated)}$ (mg g$^{-1}$) | 111.47 | 102.44 |
| | | $q_{e\,(experimental)}$ (mg g$^{-1}$) | 123.07 | 116.09 |
| | | $k_2$ (g mg$^{-1}$ min$^{-1}$) | 0.006 | 0.005 |
| | | $R^2$ | 0.981 | 0.891 |
| IPD | 6 | $C_1$ | 72.76 | 62.67 |
| | | $K_{dif1}$ (mg g$^{-1}$ Min$^{-1/2}$) | 6.93 | 6.35 |
| | | $R^2$ | 0.941 | 0.953 |
| | | $C_2$ | 109.13 | 98.35 |
| | | $K_{dif2}$ (mg g$^{-1}$ min$^{-1/2}$) | 0.04 | 0.17 |
| | | $R^2$ | 0.689 | 0.924 |

The PFO model of Lagergren [60] is described by Equation (4), where $k_1$ $(min^{-1})$ is the rate constant and $q_e$ is the highest amount of Cu (II) and Pb (II) ions eliminated at equilibrium. $k_2$ is the PSO rate constant (g/mg/min) for the PSO law (Equation (5)) [60].

$$q_t = q_e \left(1 - e^{-1k_1 t}\right) \tag{4}$$

$$q_t = \frac{t\,k_2 q_e^2}{k_2 q_e t + 1} \tag{5}$$

where $q_e$ is the adsorption capability (adsorption of the dye per unit mass of the sample, $mg\,g^{-1}$) and $q_t$ is adsorption capacity of Cu(II) and Pb(II) at time t (mg/g), respectively. $k_1$ and $k_2$ are the PFO and PSO rate constants, respectively.

The veracity (validity) of these models is evaluated using the coefficient of regression ($R^2$). Table 1 displays the parameters derived from fitting the experimental data of Cu (II) and Pb (II) adsorption to the kinetic models. As remarked, the PSO model successfully fits the kinetic adsorption data of Cu (II) and Pb (II) ions, since the value of $R^2 = 0.97$ for Cu (II) ion, and 0.99 for Pb (II).

$Cu^{2+}$ and $Pb^{2+}$ ions can be transferred from the aqueous phase to the solid phase of MgO nanosorbent through intra-particle diffusion and transport. In certain instances, intra-particular diffusion can be a limiting step in the adsorption mechanism. Intra-particular diffusion is depicted by the Weber and Morris diffusion models (Equation (6)) [61,62].

$$q_t = k_{dif} \sqrt{t} + C \tag{6}$$

Regarding the intra-particle diffusion model [63], $q_t$ is adsorption capacity of Cu(II) and Pb(II) at time t (mg/g), $k_{diff}$ is intraparticle diffusion rate constant C is the intercept that provides additional information regarding the thickness of the edge layer and $k_{dif}$ $(mg\,g^{-1}\,min^{1/2})$ is the IPD constant.

Figure 6d displays the linear plot of $q_t$ vs. $t^{1/2}$ for the adsorption of Pb (II) and Cu (II) ions onto MgO nanorods. The intraparticle diffusion rate constant $k_{dif}$ is estimated from the slope of the linear plot of $q_t$ versus $t^{1/2}$ (Figure 6d). Table 1 provides the fitting parameters of the intraparticle diffusion model. According to the fitting results, the adsorption process has two linear phases for $Cu^{2+}$ and $Pb^{2+}$. The first linear segment refers to the diffusion of $Cu^{2+}$ ($Pb^{2+}$) metal ions on the MgO nanoparticle's external surface, while the second segment represents the diffusion of $Cu^{2+}$ ($Pb^{2+}$) ions from the external surface to the pores of the MgO nanoparticle's internal structure.

### 3.4. Cu(II) and Pb(II) Adsorption Isotherms

Two nonlinear isotherm models (Freundlich and Langmuir) were employed to assess the absorption capability of MgO nanorods towards Cu(II) and Pb(II) ions. The Freundlich model is related to multilayer adsorption, while the Langmuir model is linked to a monolayer [64]. The Langmuir [64] and Freundlich [65] models are expressed by Equation (7) and Equation (8), respectively:

$$q_e = \frac{q_{max}K_L\,C_e}{1 + K_L\,C_e} \tag{7}$$

$$q_e = k_F C_e^{1/2} \tag{8}$$

where $q_m$ and $q_e$ represent the maximum adsorption capacity and the adsorption capacity at equilibrium, respectively. $K_L$ $(L \cdot mg^{-1})$ is the Langmuir constant, representing the affinity between the solute and adsorbent, and $K_F$ is the Freundlich constant.

The plots of the nonlinear Freundlich and Langmuir isotherms are illustrated in Figure 7 and the isotherms parameters are presented in Table 2. It appears that the Langmuir model fits the experimental data better than the Freundlich for Cu(II) and $Pb^{2+}$ ions because the Langmuir model has the highest $R^2$ (0.992 for Cu data and 0.997 for Pb data). This result reveals that the Cu(II) and $Pb^{2+}$ elimination by MgO nanoparticles obeyed the monolayer

adsorptions. In addition, the maximum MgO nanoparticles adsorption capacity for Cu(II) and $Pb^{2+}$ are 234.34 mg/g and 221.26 mg/g, respectively. Furthermore, the separation factor, $R_L$, might be determined using Equation (9) from the Langmuir plot [59].

$$R_L = \frac{1}{1 + K_L \, C_0} \tag{9}$$

where $R_L$ values can indicate if adsorption is favorable ($0 < R_L < 1$), unfavorable ($R_L > 1$), or irreversible ($R_L = 0$). The adsorption process of Cu(II) and Pb(II) ions over MgO nanorods is favorable since all $R_L$ values are less than a unit and more than zero.

**Table 2.** Equilibrium isotherm models for Pb(II) and Cu(II) adsorption onto MgO nanorods.

| Equilibrium Model | Equation N° | Parameters | Cu | Pb |
|---|---|---|---|---|
| Langmuir | 6 | $q_m$ (mg g$^{-1}$) | 234.34 | 221.26 |
| | | $K_L$ (mg g$^{-1}$) | 3.04 | 0.51 |
| | | $R_L$ (L mg$^{-1}$) | 0.0016 | 0.0097 |
| | | $R^2$ | 0.992 | 0.997 |
| Freundlich | 7 | $K_F$ (L mg$^{-1}$) | 177.36 | 86.56 |
| | | $R^2$ | 0.922 | 0.942 |

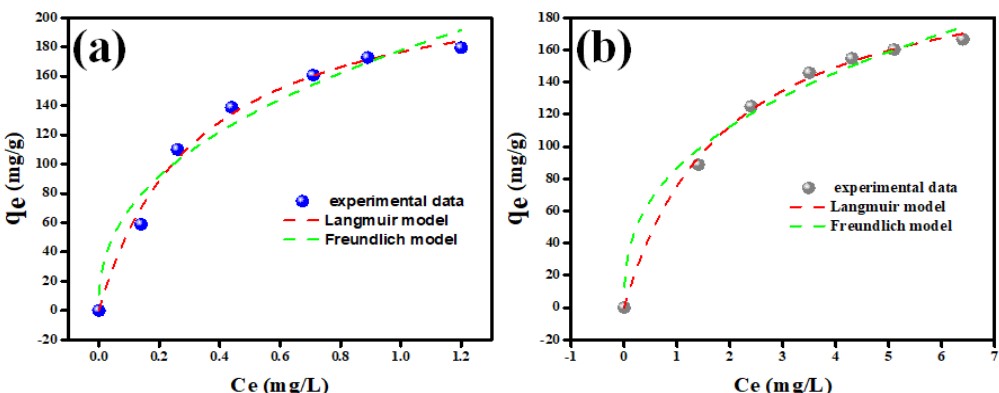

**Figure 7.** Nonlinear Cu(II) (**a**) and Pb(II) (**b**) equilibrium adsorption isotherms onto MgO nanopar–ticles fitted with Freundlich and Langmuir model.

### 3.5. Comparative Study

The maximum adsorption capacities of Cu(II) and Pb(II) ions onto MgO nanoparticles are given to be 234.34 and 221.26 mg/g, respectively. The Cu(II) adsorption capacity was higher than Pb(II), which indicates that MgO nanoparticles have a higher affinity for Cu(II) ions than $Pb^{2+}$ ions. Throughout history, numerous adsorbents have been used to eliminate Cu(II) and Pb(II) ions from wastewater. The adsorption capacities of MgO nanoparticles and a number of adsorption materials have been previously listed in Table 3. The high performance of MgO nanoparticles for the Pb(II) and Cu(II) ions elimination from an aqueous solution makes it a high and potentially effective adsorbent for the elimination of Pb(II) and Cu(II) heavy metal ions.

### 3.6. Adsorption Mechanism of Pb(II) and Cu(II)

The mechanism of Pb(II) and Cu(II) ions adsorption onto MgO may imply ion exchange, electrostatic interactions, physical adsorption, and surface complexation [74–77]. After examining the effect of pH, the optimal adsorption capacity for Pb(II) and Cu(II) ions is obtained at a pH value equal to 3. It is concluded that the electrostatic interactions between MgO and Cu(II) and Pb(II) ions may not have occurred at this pH value. Only ion exchange, physical adsorption, and surface complexation can occur. To define the mechanism of Pb(II) and Cu(II) adsorption onto nanomaterials, FTIR spectra of the MgO

nanorods after Pb(II) and Cu(II) ions adsorption were registered, as represented in Figure 8a. The OH stretching vibration band shifted significantly, as depicted in Figure 8a, indicating that lead ions engage with the OH in MgO to create the Mg–O–Pb bond. The displacement of the MgO vibration band from 867 to 861 cm$^{-1}$ confirms this result [51]. In addition, Pb–O stretching vibrations were observed at 365 cm$^{-1}$ [78], confirming the interaction of Pb(II) ions with oxygen atoms in MgO nanorods via surface complexation [74]. In addition, the observation of band at 531 cm$^{-1}$ was attributed to Cu–O vibration was suggesting well interaction between Cu(II) and MgO (Cu@MgO) [79], indicating that Cu(II) ions are adsorbed into MgO nanorods via surface complexation. Figure 8b depicts a plausible process for the adsorption of Pb(II) and Cu(II) ions into MgO nanorods.

**Table 3.** Adsorption capability of divers sorbents for the Pb(II) and Cu(II) elimination.

| Adsorbents | Metals | Adsorption Capacity (mg g$^{-1}$) | Contact Time (min) | References |
|---|---|---|---|---|
| HTDMA-modified bentonite clay | Pb(II) | 25.80 | 120 | [66] |
| KB/Zn-Fe | Pb(II) | 161.29 | 980 | [67] |
| MoS$_2$/MWCNT | Pb(II) | 90.0 | 10 | [68] |
| Orange Peels | Pb(II) | 40.05 | 120 | [10] |
| Orange Peels | Cu(II) | 38.18 | 120 | [10] |
| g-C$_3$N$_4$ | Pb(II) | 182.70 | - | [69] |
| Fe$_3$O$_4$@SiO$_2$@TiO$_2$ nanoparticles | Cu(II) | 50.50 | 30 | [70] |
| Ca$_5$(PO$_4$)$_3$OH, HA | Cu(II) | 64.14 | 480 | [71] |
| magnetic chitosan nanocomposite | Cu(II) | 12.12 | 20 | [72] |
| CPBs | Cu(II) | 169.40 | 100 | [73] |
| MgO nanoparticles | Cu(II) | 234.34 | 17 | This work |
| MgO nanoparticles | Pb(II) | 221.26 | 24 | This work |

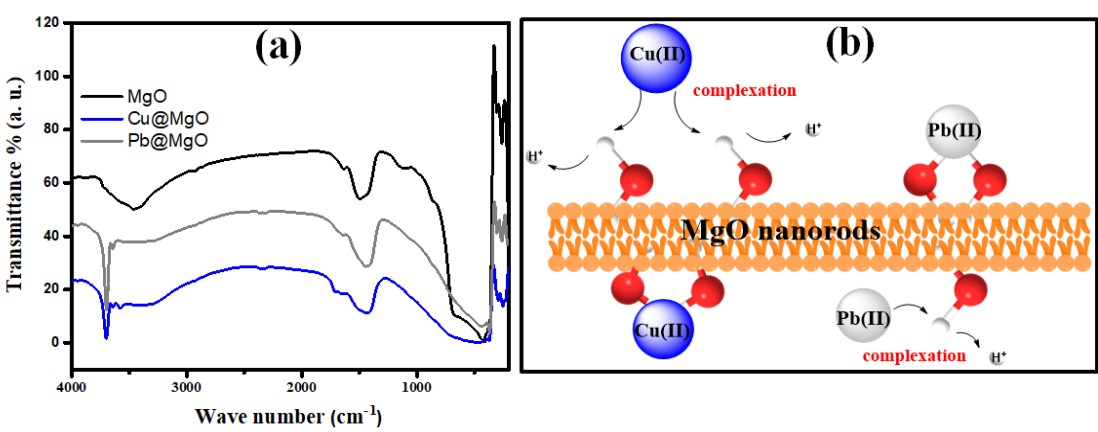

**Figure 8.** (**a**) FTIR spectra of Cu@MgO, Pb@MgO and MgO nanorods, (**b**) proposed adsorption mechanism of Cu(II) and Pb(II) ions onto MgO nanorods.

### 3.7. Regeneration Study

As is well known, the recyclability of adsorbents is a crucial aspect in determining their practical stability. Three cyclic adsorption–desorption tests were conducted using a simple alkaline solution [48]. Figure 9 shows that MgO nanorods can be well-regenerated for Cu(II) and Pb(II) ions after at least three cycles. Even on the third cycle, it was found that MgO nanorods remove about 92.6% of Cu(II) ions and about 90.3% of Pb(II) ions. The results demonstrated that MgO nanorods had good reusability.

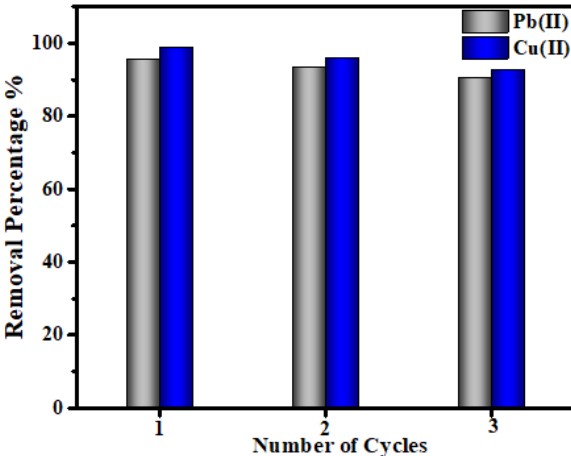

**Figure 9.** MgO nanorods recyclability.

## 4. Conclusions

In this study, MgO nanorods were effectively produced using a simple coprecipitation method. SEM analyses revealed that the MgO nanoparticles exhibit a homogeneous structure resembling a rod. The application of MgO nanorods for simultaneous removal of Cu(II) and Pb(II) from aqueous solutions was scrutinized. Several adsorption factors, including contact time, starting heavy metals concentration, interfering ions, and pH, were also investigated. The MgO nanorods present higher adsorption capacities towards Cu(II) (234.34 mg/g) and Pb(II) (221.26 mg/g) at pH values equal to 3. The elimination of Pb(II) and Cu(II) by nanocomposite reached equilibrium within 17–24 min. It was determined that the Langmuir isotherm and PSO kinetic model best fit the adsorption process, suggesting a monolayer chemisorption process. The regeneration experiments demonstrate the potential applicability of MgO nanorods for the elimination of Pb(II) and Cu(II) from aqueous solution. The results show that MgO nanorods can efficiently and rapidly remove of Cu(II) and Pb(II) heavy metals from aqueous solutions.

**Author Contributions:** Conceptualization, M.G.G. and M.A.B.A.; formal analysis and data curation, M.A.B.A.; F.A.M.A. and M.K.; writing—original draft preparation, M.G.G., M.A.B.A. and M.K. All authors have read and agreed to the published version of the manuscript.

**Funding:** This research was funded by the Deanship of Scientific Research, Imam Mohammad Ibn Saud Islamic University (IMSIU), Saudi Arabia. Grant No. (221412028).

**Data Availability Statement:** All data and information recorded or analyzed throughout this study are included in this paper.

**Acknowledgments:** The authors extend their appreciation to the Deanship of Scientific Research, Imam Mohammad Ibn Saud Islamic University (IMSIU), Saudi Arabia, for funding this research work through Grant No. (221412028).

**Conflicts of Interest:** The authors declare that they have no known competing financial interests or personal relationships that could have appeared to influence the work reported in this paper.

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
