# Peer review of "Efficient and Rapid Removal of Pb(II) and Cu(II) Heavy Metals from Aqueous Solutions by MgO Nanorods"

_inorganics, doi:10.3390/inorganics10120256_

Round 1
Reviewer 1 Report
Manuscript: Efficiently and Rapidly Remove Pb (II) and Cu (II) Heavy Metals from Aqueous Solutions by MgO Nanorods
In presented study authors investigated the adsorption capability of MgO nanorods for the quick and effective elimination of Cu (II) and Pb (II) heavy metals from wastewater. Although study lacks of novelty, authors generally did good job and most of the experiments were well-designed. The main conclusion is that Manuscript should be significantly improved before the acceptance for the publication.
1. Abstract:
Put “via* simple” instead of “via a simple”. *in Italic
I think as-prepared is more appropriate word in comparison to as-obtained you used.
2. The Introduction section lacks of relevant literature citations. One or two references after some statements are not enough. Please, correct it to emphasize the significance of your topic.
3. Section 2.4: Please put the legends below equations 1 and 2 explaining symbols given in them.
4. The title of 3.1 section: Replace “charecteristisation” with “characterization”.
5. FTIR: If you started with irreducible representation (A2u), symmetry of every vibration should be assigned.
6. The sentence “The generated nanomaterials' elemental mapping showed that O and Mg were evenly dispersed within the MgO nanoparticles (Figure 3 d, e).” does not make a sense since you obtained pure MgO (1:1) phase.
7. “On the other hand, Transmission electron microscopic (TEM) micrograph (Fig. 2f) of the prepared MgO confirm the formation of nanorods of MgO with average diameter 10.7 nm and length of 156 nm with high homogeneity”.
From obtained micrograph I could not see what authors claimed since they did not measure nanorods. Also, I’m afraid that presented micrograph is not good choice for it. Please, provide micrograph of better resolution and higher magnification with measurements.
8. Section 3.2.2. What did you use to set desired pH value? HCl and NaOH? Please mention it in the Experimental section. Also, such acidic environment (pH = 1) is not reasonable choice for pH dependence experiments since it is well-known fact about MgO behavior in acidic conditions i.e., dissolution occurs. This fact explains low adsorption performance at pH 1 and 2.
9. To improve quality of your study, provide results of reusability experiments i.e., adsorption efficiency after several adsorption/desorption cycles. Authors are also encouraged to provide results of their adsorbent performance in the presence of interfering ions.
10. Authors could not claim "The application of MgO nanorods for simultaneous removal of Cu (II) and Pb(II) from wastewater was scrutinized" since they did not do experiments with real wastewater. So "aqueous solution" is more appropriate word since solutions containing Pb and Cu ions were used in all experiments.
Reviewer 2 Report
Dear authors,
Attached you will find the mandatory points raised. The topic is very interesting, however, extensive restructuring and discussion is necessary before its possible publication.

Round 2
Reviewer 1 Report
Authors should correct the title of the manuscript to Efficient and Rapid Removal of Pb(II) and Cu(II) Heavy Metals from Aqueous Solutions by MgO Nanorods.
Delete the results obtained for pH = 1 and pH = 9 in Fig4b. No sense in using these values.
After these corrections paper can be accepted for publication.
Reviewer 2 Report
Dear Authors,
Thank you for considering my comments and suggestions. I recommend the publication of the revised version.
Author Response
Thanks for the feedback and for sending me your comments.